# Strategizing against No-regret Learners

**Yuan Deng**
Duke University
ericdy@cs.duke.edu

**Jon Schneider**
Google Research
jschnei@google.com

**Balasubramanian Sivan**
Google Research
balusivan@google.com

## Abstract

How should a player who repeatedly plays a game against a no-regret learner strategize to maximize his utility? We study this question and show that under some mild assumptions, the player can always guarantee himself a utility of at least what he would get in a Stackelberg equilibrium of the game. When the no-regret learner has only two actions, we show that the player cannot get any higher utility than the Stackelberg equilibrium utility. But when the no-regret learner has more than two actions and plays a mean-based no-regret strategy, we show that the player can get strictly higher than the Stackelberg equilibrium utility. We provide a characterization of the optimal game-play for the player against a mean-based no-regret learner as a solution to a control problem. When the no-regret learner's strategy also guarantees him a no-swap regret, we show that the player cannot get anything higher than a Stackelberg equilibrium utility.

## 1  Introduction

Consider a two player bimatrix game with a finite number of actions for each player repeated over $T$ rounds. When playing a repeated game, a widely adopted strategy is to employ a *no-regret learning algorithm*: a strategy that guarantees the player that in hindsight no single action when played throughout the game would have performed significantly better. Knowing that one of the players (the *learner*) is playing a no-regret learning strategy, what is the optimal gameplay for the other player (the *optimizer*)? This question is the focus of our work.

If this were a single-shot strategic game where learning is not relevant, a (pure or mixed strategy) Nash equilibrium is a reasonable prediction of the game's outcome. In the $T$ rounds game with learning, can the optimizer guarantee himself a per-round utility of at least what he could get in a single-shot game? Is it possible to get significantly more utility than this? Does this utility depend on the specific choice of learning algorithm of the learner? What gameplay the optimizer should adopt to achieve maximal utility? None of these questions are straightforward, and indeed none of these have unconditional answers.

**Our results.**  Central to our results is the idea of the Stackelberg equilibrium of the underlying game. The Stackelberg variant of our game is a single-shot two-stage game where the optimizer is the first player and can publicly commit to a mixed strategy; the learner then best responds to this strategy. The Stackelberg equilibrium is the resulting equilibrium of this game when both players play optimally. Note that the optimizer's utility in the Stackelberg equilibrium is always weakly larger than his utility in any (pure or mixed strategy) Nash equilibrium, and is often strictly larger.

Let $V$ be the utility of the optimizer in the Stackelberg equilibrium. With some mild assumptions on the game, we show that the optimizer can always guarantee himself a utility of at least $(V-\varepsilon)T-o(T)$ in $T$ rounds for any $\varepsilon > 0$, irrespective of the learning algorithm used by the learner as long as it has the no-regret guarantee (Theorem 4). This means that if one of the players is a learner the other player can already profit over the Nash equilibrium regardless of the specifics of the learning algorithm employed or the structure of the game. Further, if any one of the following conditions is true:

1. the game is a constant-sum game,
2. the learner's no-regret algorithm has the stronger guarantee of no-swap regret (see Section 2),
3. the learner has only two possible actions in the game,

the optimizer cannot get a utility higher than $VT + o(T)$ (see Theorem 5, Theorem 6, Theorem 7).

If the learner employs a learning algorithm from a natural class of algorithms called mean-based learning algorithms [Braverman et al., 2018] (see Section 2) that includes popular no-regret algorithms like the Multiplicative Weights algorithm, the Follow-the-Perturbed-Leader algorithm, and the EXP3 algorithm, we show that there exist games where the optimizer can guarantee himself a utility $V'T - o(T)$ for some $V' > V$ (see Theorem 8). We note the contrast between the cases of 2 and 3 actions for the learner: in the 2-actions case even if the learner plays a mean-based strategy, the optimizer cannot get anything more than $VT + o(T)$ (Theorem 7), whereas with 3 actions, there are games where he is able to guarantee a linearly higher utility.

Given this possibility of exceeding Stackelberg utility, our final result is on the nature and structure of the *utility optimal gameplay* for the optimizer against a learner that employs a mean-based strategy. First, we give a crisp characterization of the optimizer's asymptotic optimal algorithm as the solution to a control problem (see Section 4.2) in $N$ dimensions where $N$ is the number of actions for the learner. This characterization is predicated on the fact that just knowing the cumulative historical utilities of each of the learner's actions is essentially enough information to accurately predict the learner's next action in the case of a mean-based learner. These $N$ cumulative utilites thus form an $N$-dimensional "state" for the learner which the optimizer can manipulate via their choice of action. We then proceed to make multiple observations that simplify the solution space for this control problem. We leave as a very interesting open question of computing or characterizing the optimal solution to this control problem and we further provide one conjecture of a potential characterization.

**Comparison to prior work.** The very recent work of Braverman et al. [2018] is the closest to ours. They study the specific 2-player game of an auction between a single seller and single buyer. The main difference from Braverman et al. [2018] is that they consider a Bayesian setting where the buyer's type is drawn from a distribution, whereas there is no Bayesian element in our setting. But beyond that the seller's choice of the auction represents his action, and the buyer's bid represents her action. They show that regardless of the specific algorithm used by the buyer, as long as the buyer plays a no-regret learning algorithm the seller can always earn at least the optimal revenue in a single shot auction. Our Theorem 4 is a direct generalization of this result to arbitrary games without any structure. Further Braverman et al. [2018] show that there exist no-regret strategies for the buyer that guarantee that the seller cannot get anything better than the single-shot optimal revenue. Our Theorems 5, 6 and 7 are both generalizations and refinements of this result, as they pinpoint both the exact learner's strategies and the kind of games that prevent the optimizer from going beyond the Stackelberg utility. Braverman et al. [2018] show that when the buyer plays a mean-based strategy, the seller can design an auction to guarantee him a revenue beyond the per round auction revenue. Our control problem can be seen as a rough parallel and generalization of this result.

**Other related work.** The first notion of regret (without the swap qualification) we use in the paper is also referred to as external-regret (see Hannan [1957], Foster and Vohra [1993], Littlestone and Warmuth [1994], Freund and Schapire [1997], Freund and Schapire [1999], Cesa-Bianchi et al. [1997]). The other notion of regret we use is swap regret. There is a slightly weaker notion of regret called internal regret that was defined earlier in Foster and Vohra [1998], which allows all occurrences of a given action $x$ to be replaced by another action $y$. Many no-internal-regret algorithms have been designed (see for example Hart and Mas-Colell [2000], Foster and Vohra [1997, 1998, 1999], Cesa-Bianchi and Lugosi [2003]). The stronger notion of swap regret was introduced in Blum and Mansour [2005], and it allows one to simultaneously swap several pairs of actions. Blum and Mansour show how to efficiently convert a no-regret algorithm to a no-swap-regret algorithm. One of the reasons behind the importance of internal and swap regret is their close connection to the central notion of correlated equilibrium introduced by Aumann [1974]. In a general $n$ players game, a distribution over action profiles of all the players is a correlated equilibrium if every player has zero internal regret. When all players use algorithms with no-internal-regret guarantees, the time averaged strategies of the players converges to a correlated equilibrium (see Hart and Mas-Colell [2000]). When all players simply use algorithms with no-external-regret guarantees, the time averaged strategies of the players converges to the weaker notion of coarse correlated equilibrium. When the game is a zero-sum game,

the time-averaged strategies of players employing no-external-regret dynamics converges to the Nash equilbrium of the game.

On the topic of optimizing against a no-regret-learner, Agrawal et al. [2018] study a setting similar to Braverman et al. [2018] but also consider other types of buyer behavior apart from learning, and show to how to robustly optimize against various buyer strategies in an auction.

## 2 Model and Preliminaries

### 2.1 Games and equilibria

Throughout this paper, we restrict our attention to simultaneous two-player bimatrix games $G$. We refer to the first player as the *optimizer* and the second player as the *learner*. We denote the set of actions available to the optimizer as $\mathcal{A} = \{a_1, a_2, \ldots, a_M\}$ and the set of actions available to the learner as $\mathcal{B} = \{b_1, b_2, \ldots, b_N\}$. If the optimizer chooses action $a_i$ and the learner chooses action $b_j$, then the optimizer receives utility $u_O(a_i, b_j)$ and the learner receives utility $u_L(a_i, b_j)$. We normalize the utility such that $|u_O(a_i, b_j)| \leq 1$ and $|u_L(a_i, b_j)| \leq 1$. We write $\Delta(\mathcal{A})$ and $\Delta(\mathcal{B})$ to denote the set of mixed strategies for the optimizer and learner respectively. When the optimizer plays $\alpha \in \Delta(\mathcal{A})$ and the learner plays $\beta \in \Delta(\mathcal{B})$, the optimizer's utility is denoted by $u_O(\alpha, \beta) = \sum_{i=1}^{M} \sum_{j=1}^{N} \alpha_i \beta_j u_O(a_i, b_j)$, similarly for the learner's utility.

We say that a strategy $b \in \mathcal{B}$ is a best-response to a strategy $\alpha \in \Delta(\mathcal{A})$ if $b \in \operatorname{argmax}_{b' \in \mathcal{B}} u_L(\alpha, b')$. We are now ready to define Stackelberg equilibrium [Von Stackelberg, 2010].

**Definition 1.** *The* Stackelberg equilibrium *of a game is a pair of strategies $\alpha \in \Delta(\mathcal{A})$ and $b \in \mathcal{B}$ that maximizes $u_O(\alpha, b)$ under the constraint that $b$ is a best-response to $\alpha$. We call the value $u_O(\alpha, b)$ the* Stackelberg value *of the game.*

A game is *zero-sum* if $u_O(a_i, b_j) + u_L(a_i, b_j) = 0$ for all $i \in [M]$ and $j \in [N]$; likewise, a game is *constant-sum* if $u_O(a_i, b_j) + u_L(a_i, b_j) = C$ for some fixed constant $C$ for all $i \in [M]$ and $j \in [N]$. Note that for zero-sum or constant-sum games, the Stackelberg equilibrium coincides with the standard notion of Nash equilibrium due to the celebrated minimax theorem [von Neumann, 1928]. Moreover, throughout this paper, we assume that the learner does not have weakly dominated strategies: a strategy $b \in \mathcal{B}$ is weakly dominated if there exists $\beta \in \Delta(\mathcal{B} \setminus \{b\})$ such that for all $a \in \mathcal{A}$, $u_L(a, \beta) \geq u_L(a, b)$.

We are interested in the setting where the optimizer and the learner repeatedly play the game $G$ for $T$ rounds. We will denote the optimizer's action at time $t$ as $a^t$; likewise we will denote the learner's action at time $t$ as $b^t$. Both the optimizer and learner's utilities are additive over rounds with no discounting.

The optimizer's strategy can be adaptive (i.e. $a^t$ can depend on the previous values of $b^t$) or non-adaptive (in which case it can be expressed as a sequence of mixed strategies $(\alpha^1, \alpha^2, \ldots, \alpha^T)$). Unless otherwise specified, all positive results (results guaranteeing the optimizer can guarantee some utility) apply for non-adaptive optimizers and all negative results apply even to adaptive optimizers. As the name suggests, the learner's (adaptive) strategy will be specified by some variant of a low-regret learning algorithm, as described in the next section.

### 2.2 No-regret learning and mean-based learning

In the classic multi-armed bandit problem with $T$ rounds, the learner selects one of $K$ options (a.k.a. arms) on round $t$ and receives a reward $r_{i,t} \in [0, 1]$ if he selects option $i$. The rewards can be chosen adversarially and the learner's objective is to maximize her total reward.

Let $i_t$ be the arm pulled by the learner at round $t$. The *regret* for a (possibly randomized) learning algorithm Alg is defined as the difference between performance of the algorithm Alg and the best arm: $\mathrm{Reg}(\mathrm{Alg}) = \max_i \sum_{t=1}^{T} r_{i,t} - r_{i_t,t}$. An algorithm Alg for the multi-armed bandit problem is *no-regret* if the expected regret is sub-linear in $T$, i.e., $\mathbb{E}[\mathrm{Reg}(\mathrm{Alg})] = o(T)$. In addition to the *bandits* setting in which the learner only learns the reward of the arm he pulls, our results also apply to the *experts* setting in which the learner can learn the rewards of all arms for every round. Simple no-regret strategies exist in both the bandits and the experts settings.

Among no-regret algorithms, we are interested in two special classes of algorithms. The first is the class of *mean-based* strategies:

**Definition 2** (Mean-based Algorithm). *Let $\sigma_{i,t} = \sum_{s=1}^{t} r_{i,s}$ be the cumulative reward for pulling arm $i$ for the first $t$ rounds. An algorithm is $\gamma$-mean-based if whenever $\sigma_{i,t} < \sigma_{j,t} - \gamma T$, the probability for the algorithm to pull arm $i$ on round $t$ is at most $\gamma$. An algorithm is mean-based if it is $\gamma$-mean-based for some $\gamma = o(1)$.*

Intuitively, mean-based strategies are strategies that play the arm that historically performs the best. Braverman et al. [2018] show that many no-regret algorithms are mean-based, including commonly used variants of EXP3 (for the bandits setting), the Multiplicative Weights algorithm (for the experts setting) and the Follow-the-Perturbed-Leader algorithm (for the experts setting).

The second class is the class of *no-swap-regret* algorithms:

**Definition 3** (No-Swap-Regret Algorithm). *The* swap regret $\text{Reg}_{swap}(\text{Alg})$ *of an algorithm* Alg *is defined as*

$$\text{Reg}_{swap}(\text{Alg}) = \max_{\pi:[K] \to [K]} \text{Reg}(\text{Alg}, \pi) = \sum_{t=1}^{T} r_{\pi(i_t),t} - r_{i_t,t}$$

*where the maximum is over all functions $\pi$ mapping options to options. An algorithm is* no-swap-regret *if the expected swap regret is sublinear in $T$, i.e. $\mathbb{E}[\text{Reg}_{swap}(\text{Alg})] = o(T)$.*

Intuitively, no-swap-regret strategies strengthen the no-regret criterion in the following way: no-regret guarantees the learning algorithm performs as well as the best possible arm overall, but no-swap-regret guarantees the learning algorithm performs as well as the best possible arm over each subset of rounds where the same action is played. Given a no-regret algorithm, a no-swap-regret algorithm can be constructed via a clever reduction (see Blum and Mansour [2005]).

## 3 Playing against no-regret learners

### 3.1 Achieving Stackelberg equilibrium utility

To begin with, we show that the optimizer can achieve an average utility per round arbitrarily close to the Stackelberg value against a no-regret learner.

**Theorem 4.** *Let $V$ be the Stackelberg value of the game $G$. If the learner is playing a no-regret learning algorithm, then for any $\varepsilon > 0$, the optimizer can guarantee at least $(V - \varepsilon)T - o(T)$ utility.*

*Proof.* Let $(\alpha, b)$ be the Stackelberg equilibrium of the game $G$. Since $(\alpha, b)$ forms a Stackelberg equilibrium, $b \in \text{argmax}_{b'} u_L(\alpha, b')$. Moreover, by the assumption that the learner does not have a weakly dominated strategy, there does not exist $\beta \in \Delta(\mathcal{B} \setminus \{b\})$ such that for all $a \in \mathcal{A}$, $u_L(a, \beta) \geq u_L(a, b)$. By Farkas's lemma [Farkas, 1902], there must exist an $\alpha' \in \Delta(\mathcal{A})$ such that for all $b' \in \mathcal{B} \setminus \{b\}$, $u_L(\alpha', b) \geq u_L(\alpha', b') + \kappa$ for $\kappa > 0$.

Therefore, for any $\delta \in (0, 1)$, the optimizer can play the strategy $\alpha^* = (1 - \delta)\alpha + \delta\alpha'$ such that $b$ is the unique best response to $\alpha^*$ and playing strategy $b' \neq b$ will induce a utility loss at least $\delta\kappa$ for the learner. As a result, since the leaner is playing a no-regret learning algorithm, in expectation, there is at most $o(T)$ rounds in which the learner plays $b' \neq b$. It follows that the optimizer's utility is at least $VT - \delta(V - u_O(\alpha', b))T - o(T)$. Thus, we can conclude our proof by setting $\varepsilon = \delta(V - u_O(\alpha', b))$. $\square$

Next, we show that in the special class of constant-sum games, the Stackelberg value is the best that the optimizer can hope for when playing against a no-regret learner.

**Theorem 5.** *Let $G$ be a constant-sum game, and let $V$ be the Stackelberg value of this game. If the learner is playing a no-regret algorithm, then the optimizer receives no more than $VT + o(T)$ utility.*

*Proof.* Let $\vec{a} = (a^1, \cdots, a^T)$ be the sequence of the optimizer's actions. Moreover, let $\alpha^* \in \Delta(\mathcal{A})$ be a mixed strategy such that $\alpha^*$ plays $a_i \in \mathcal{A}$ with probability $\alpha_i^* = |\{t \mid a^t = a_i\}|/T$.

Since the learner is playing a no-regret learning algorithm, the learner's cumulative utility is at least $\max_{b' \in \mathcal{B}} u_L(a^*, b')T - o(T) = CT - (\min_{b' \in \mathcal{B}} u_O(a^*, b')T + o(T))$, where $C$ is the constant sum, which implies that the optimizer's utility is at most

$$\max_{a^* \in \Delta(\mathcal{A})} \min_{b' \in \mathcal{B}} u_O(a^*, b')T + o(T) = VT + o(T)$$

where the equality follows that the Stackelberg value is equal to the minimax value by the minimax theorem for a constant-sum game. □

## 3.2 No-swap-regret learning

In this section, we show that if the learner is playing a no-swap-regret algorithm, the optimizer can only achieve their Stackelberg utility per round.

**Theorem 6.** *Let $V$ be the Stackelberg value of the game $G$. If the learner is playing a no-swap-regret algorithm, then the optimizer will receive no more than $VT + o(T)$ utility.*

*Proof.* Let $\vec{a} = (a^1, \cdots, a^T)$ be the sequence of the optimizer's actions and let $\vec{b} = (b^1, \cdots, b^T)$ be the realization of the sequence of the learner's actions. Moreover, let $\Pr[\vec{b}]$ be the probability that the learner (who is playing some no-swap-regret learning algorithm) plays $\vec{b}$ given that the adversary plays $\vec{a}$. Then, the marginal probability for the learner to play $b_j \in \mathcal{B}$ at round $t$ is

$$\Pr[b^t = b_j] = \sum_{\vec{b}: b^t = b_j} \Pr[\vec{b}].$$

Let $\alpha^{b_j} \in \Delta(\mathcal{A})$ be a mixed strategy such that $\alpha^{b_j}$ plays $a_i \in \mathcal{A}$ with probability

$$\alpha_i^{b_j} = \frac{\sum_{t:a^t = a_i} \Pr[b^t = b_j]}{\sum_t \Pr[b^t = b_j]}.$$

Let $\bar{\mathcal{B}} = \{b_j \in \mathcal{B} : b_j \notin \text{argmax}_{b'} u_L(\alpha^{b_j}, b')\}$ and consider a mapping $\pi$ such that $\pi(b_j) \in \text{argmax}_{b'} u_L(\alpha^{b_j}, b')$. Then, the swap-regret under $\pi$ is

$$\sum_{b_j \in \mathcal{B}} \left( \left( u_L(\alpha^{b_j}, \pi(b_j)) - u_L(\alpha^{b_j}, b_j) \right) \cdot \sum_t \Pr[b^t = b_j] \right)$$

$$= \sum_{b_j \in \bar{\mathcal{B}}} \left( \left( u_L(\alpha^{b_j}, \pi(b_j)) - u_L(\alpha^{b_j}, b_j) \right) \cdot \sum_t \Pr[b^t = b_j] \right)$$

$$\geq \delta \cdot \sum_{b_j \in \bar{\mathcal{B}}} \left( \sum_t \Pr[b^t = b_j] \right)$$

where $\delta = \min_{b_j \in \bar{\mathcal{B}}} \left( u_L(\alpha^{b_j}, \pi(b_j)) - u_L(\alpha^{b_j}, b_j) \right)$. Therefore, since the learner is playing a no-swap-regret algorithm, we have $\sum_{b_j \in \bar{\mathcal{B}}} \left( \sum_t \Pr[b^t = b_j] \right) = o(T)$.

Moreover, for $b_j \in \mathcal{B} \setminus \bar{\mathcal{B}}$, the optimizer's utility when the learner plays $b_j$ is at most

$$u_O(\alpha^{b_j}, b_j) \cdot \sum_t \Pr[b^t = b_j] \leq V \cdot \sum_t \Pr[b^t = b_j].$$

Thus, the optimizer's utility is at most

$$\sum_{b_j \in \mathcal{B}} \left( u_O(\alpha^{b_j}, b_j) \cdot \sum_t \Pr[b^t = b_j] \right)$$

$$= \sum_{b_j \in \mathcal{B} \setminus \bar{\mathcal{B}}} \left( u_O(\alpha^{b_j}, b_j) \cdot \sum_t \Pr[b^t = b_j] \right) + \sum_{b_j \in \bar{\mathcal{B}}} \left( u_O(\alpha^{b_j}, b_j) \cdot \sum_t \Pr[b^t = b_j] \right)$$

$$\leq V \cdot \sum_{b_j \in \mathcal{B} \setminus \bar{\mathcal{B}}} \left( \sum_t \Pr[b^t = b_j] \right) + 1 \cdot \sum_{b_j \in \bar{\mathcal{B}}} \left( \sum_t \Pr[b^t = b_j] \right)$$

$$\leq VT + o(T).$$

$\square$

**Theorem 7.** *Let $G$ be a game where the learner has $N = 2$ actions, and let $V$ be the Stackelberg value of this game. If the learner is playing a no-regret algorithm, then the optimizer receives no more than $VT + o(T)$ utility.*

*Proof.* By Theorem 6, it suffices to show that when there are two actions for the learner, a no-regret learning algorithm is in fact a no-swap-regret learning algorithm.

When there are only two actions, there are three possible mappings from $\mathcal{B} \to \mathcal{B}$ other than the identity mapping. Let $\pi^1$ be a mapping such that $\pi^1(b_1) = b_1$ and $\pi^1(b_2) = b_1$, $\pi^2$ be a mapping such that $\pi^2(b_1) = b_2$ and $\pi^2(b_2) = b_2$, and $\pi^3$ be a mapping such that $\pi^3(b_1) = b_2$ and $\pi^3(b_2) = b_1$.

Since the learner is playing a no-regret learning algorithm, we have $\mathbb{E}[\text{Reg}(\mathcal{A}, \pi^1)] = o(T)$ and $\mathbb{E}[\text{Reg}(\mathcal{A}, \pi^2)] = o(T)$. Moreover, notice that

$$\mathbb{E}[\text{Reg}(\mathcal{A}, \pi^3)] = \mathbb{E}[\text{Reg}(\mathcal{A}, \pi^1)] + \mathbb{E}[\text{Reg}(\mathcal{A}, \pi^2)] = o(T),$$

which concludes the proof. $\square$

## 4 Playing against mean-based learners

From the results of the previous section, it is natural to conjecture that no optimizer can achieve more than the Stackelberg value per round if playing against a no-regret algorithm. After all, this is true for the subclass of no-swap-regret algorithms (Theorem 6) and is true for simple games: constant-sum games (Theorems 5) and games in which the learner only has two actions (Theorem 7).

In this section we show that this is *not* the case. Specifically, we show that there exist games $G$ where an optimizer can win strictly more than the Stackelberg value every round when playing against a mean-based learner. We emphasize that the same strategy for the optimizer will work against *any* mean-based learning algorithm the learner uses.

We then proceed to characterize the optimal strategy for a non-adaptive optimizer playing against a mean-based learner as the solution to an optimal control problem in $N$ dimensions (where $N$ is the number of actions of the learner), and make several preliminary observations about structure an optimal solution to this control problem must possess. Understanding how to efficiently solve this control problem (or whether the optimal solution is even computable) is an intriguing open question.

### 4.1 Beating the Stackelberg value

We begin by showing it is possible for the optimizer to get significantly (linear in $T$) more utility when playing against a mean-based learner.

**Theorem 8.** *There exists a game $G$ with Stackelberg value $V$ where the optimizer can receive utility at least $V'T - o(T)$ against a mean-based learner for some $V' > V$.*

*Proof.* Assume that the learner is using a $\gamma$-mean-based algorithm. Consider the bimatrix game shown in Table 1 in which the optimizer is the row player (These utilities are bounded in $[-2, 2]$ instead of $[-1, 1]$ for convenience; we can divide through by 2 to get a similar example where utility is bounded in $[-1, 1]$). We first argue that the Stackelberg value of this game is 0. Notice that if the optimizer plays Bottom with probability more than $0.5$, then the learner's best response is to play Mid, resulting in a $-2$ utility for the optimizer . However, if the optimizer plays Bottom with probability at most $0.5$, the expected utility for the optimizer from each column is at most 0. Therefore, in the Stackelberg equilibrium, the optimizer will play Top and Bottom with probability $0.5$ each, and the learner will best respond with purely playing Right.

|        | Left | Mid | Right |
|--------|------|-----|-------|
| Top    | $(0, \sqrt{\gamma})$ | (-2, -1) | (-2, 0) |
| Bottom | (0, -1) | (-2, 1) | (2, 0) |

Table 1: Example game for beating the Stackelberg value.

However, the optimizer can obtain utility $T - o(T)$ by playing Top for the first $\frac{1}{2}T$ rounds and then playing Bottom for the remaining $\frac{1}{2}T$ rounds. Given the optimizer's strategy, for the first $\frac{1}{2}T$ rounds, the learner will play Left with probability at least $(1 - 2\gamma)$ after first $\sqrt{\gamma}T$ rounds. For the remaining $\frac{1}{2}T$ rounds, the learner will switch to play Right with probability at least $(1 - 2\gamma)$ between $(\frac{1+\sqrt{\gamma}}{2} + \gamma)T$-th round and $(1 - \gamma)T$-th round, since the cumulative utility for playing Left is at most $\frac{1}{2}T \cdot \sqrt{\gamma} - \frac{\sqrt{\gamma}}{2}T - \gamma T = -\gamma T$ and the cumulative utility for playing Mid is at most $-\gamma T$.

Therefore, the cumulative utility for the optimizer for the first $\frac{1}{2}T$ rounds is at least

$$(1 - 2\gamma)(\frac{1}{2} - \sqrt{\gamma})T \cdot 0 + \left( \frac{1}{2}T - (1 - 2\gamma)(\frac{1}{2} - \sqrt{\gamma})T \right) \cdot (-2) = -o(T),$$

and the cumulative utility for the optimizer for the remaining $\frac{1}{2}T$ rounds is at least

$$(1 - 2\gamma)(\frac{1}{2} - \frac{\sqrt{\gamma}}{2} - 2\gamma)T \cdot 2 + \left( \frac{1}{2}T - (1 - 2\gamma)(\frac{1}{2} - \frac{\sqrt{\gamma}}{2} - 2\gamma)T \right) \cdot (-2) = T - o(T).$$

Thus, the optimizer can obtain a total utility $T - o(T)$, which is greater than $VT = 0$ for the Stackelberg value $V = 0$ in this game. $\qquad \square$

## 4.2 The geometry of mean-based learning

We have just seen that it is possible for the optimizer to get more than the Stackelberg value when playing against a mean-based learner. This raises an obvious next question: how much utility can an optimizer obtain when playing against a mean-based learner? What is the largest $\alpha$ such that an optimizer can always obtain utility $\alpha T - o(T)$ against a mean-based learner?

In this section, we will see how to reduce the problem of constructing the optimal gameplay of a non-adaptive optimizer to solving a control problem in $N$ dimensions. The primary insight is that a mean-based learner's behavior depends only on their historical cumulative utilities for each of their $N$ actions, and therefore we can characterize the essential "state" of the learner by a tuple of $N$ real numbers that represent the cumulative utilities for different actions. The optimizer can control the state of the learner by playing different actions, and in different regions of the state space the learner plays specific responses.

More formally, our control problem will involve constructing a path in $\mathbb{R}^N$ starting at the origin. For each $i \in [N]$, let $S_i$ equals the subset of $(u_1, u_2, \ldots, u_N) \in \mathbb{R}^N$ where $u_i = \max(u_1, u_2, \ldots, u_N)$ (this will represent the subset of state space where the learner will play action $b_i$). Note that these sets $S_i$ (up to some intersection of measure 0) partition the entire space $\mathbb{R}^N$.

We represent the optimizer's strategy $\pi$ as a sequence of tuples $(\alpha_1, t_1), (\alpha_2, t_2), \ldots, (\alpha_k, t_k)$ with $\alpha_i \in \Delta(\mathcal{A})$ and $t_i \in [0, 1]$ satisfying $\sum_i t_i = 1$. Here the tuple $(\alpha_i, t_i)$ represents the optimizer playing mixed strategy $\alpha_i$ for a $t_i$ fraction of the total rounds. This strategy evolves the learner's state as follows. The learner originally starts at the state $P_0 = 0$. After the $i$th tuple $(\alpha_i, t_i)$, the learner's state evolves according to $P_i = P_{i-1} + t_i(u_L(\alpha_i, b_1), u_L(\alpha_i, b_2), \ldots, u_L(\alpha_i, b_N))$ (in fact, the state linearly interpolates between $P_{i-1}$ and $P_i$ as the optimizer plays this action). For simplicity, we will assume that positive combinations of vectors of the form $(u_L(\alpha_i, b_1), u_L(\alpha_i, b_2), \ldots, u_L(\alpha_i, b_N))$ can generate the entire state space $\mathbb{R}^N$.

To characterize the optimizer's reward, we must know which set $S_i$ the learner's state belongs to. For this reason, we will insist that for each $1 \le i \le k$, there exists a $j_i$ such that both $P_{i-1}$ and $P_i$ belong to the same region $S_{j_i}$. It is possible to convert any strategy $\pi$ into a strategy of this form by subdividing a step $(\alpha, t)$ that crosses a region boundary into two steps $(\alpha, t')$ and $(\alpha, t'')$ with $t = t' + t''$ so that the first step stops exactly at the region boundary. If there is more than one possible choice for $j_i$ (i.e. $P_{i-1}$ and $P_i$ lie on the same region boundary), then without loss of generality we let the optimizer choose $j_i$, since the optimizer can always modify the initial path slightly so that $P_i$ and $P_{i-1}$ both lie in a unique region.

Once we have done this, the optimizer's average utility per round is given by the expression:

$$U(\pi) = \sum_{i=1}^{k} t_i u_O(\alpha_i, b_{j_i}).$$

**Theorem 9.** *Let $U^* = \sup_\pi U(\pi)$ where the supremum is over all valid strategies $\pi$ in this control game. Then*

1. *For any $\varepsilon > 0$, there exists a non-adaptive strategy for the optimizer which guarantees expected utility at least $(U^* - \varepsilon)T - o(T)$ when playing against any mean-based learner.*

2. *For any $\varepsilon > 0$, there exists no non-adaptive strategy for the optimizer which can guarantee expected utility at least $(U^* + \varepsilon)T + o(T)$ when playing against any mean-based learner.*

Understanding how to solve this control problem (even inefficiently, in finite time) is an interesting open problem. In the remainder of this section, we make some general observations which will let us cut down the strategy space of the optimizer even further and propose a conjecture to the form of the optimal strategy.

The first observation is that when the learner has $N$ actions, our state space is truly $N - 1$ dimensional, not $N$ dimensional. This is because in addition to the learner's actions only depending on the cumulative reward for each action, they in fact only depend on the differences between cumulative rewards for different actions (see Definition 2). This means we can represent the state of the learner as a vector $(x_1, x_2, \ldots, x_{N-1}) \in \mathbb{R}^{N-1}$, where $x_i = u_i - u_N$. The sets $S_i$ for $1 \le i \le N - 1$ can be written in terms of the $x_i$ as

$$S_i = \{x | x_i = \max(x_1, \ldots, x_{N-1}, 0)\}$$

and

$$S_N = \{x | 0 = \max(x_1, \ldots, x_{N-1}, 0)\}.$$

The next observation is that if the optimizer makes several consecutive steps in the same region $S_i$, we can combine them into a single step. Specifically, assume $P_i$, $P_{i+1}$, and $P_{i+2}$ all belong to some region $S_j$, where $(\alpha_i, t_i)$ sends $P_i$ to $P_{i+1}$ and $(\alpha_{i+1}, t_{i+1})$ sends $P_{i+1}$ to $P_{i+2}$. Then replacing these two steps with $\left( \frac{\alpha_i t_i + \alpha_{i+1} t_{i+1}}{t_i + t_{i+1}}, t_i + t_{i+1} \right)$ results in a strategy with the exact same reward $U(\pi)$. Applying this fact whenever possible, this means we can restrict our attention to strategies where all $P_i$ (with the possible exception of the final state $P_k$) lie on the boundary of two or more regions $S_i$.

Finally, we observe that this control problem is scale-invariant; if

$$\pi = ((\alpha_1, t_1), (\alpha_2, t_2), \ldots, (\alpha_n, t_n))$$

is a valid policy that obtains utility $U$, then

$$\lambda \pi = ((\alpha_1, \lambda t_1), (\alpha_2, \lambda t_2), \ldots, (\alpha_n, \lambda t_n))$$

is another valid policy (with the exception that $\sum t_i = \lambda$, not 1) which obtains utility $\lambda U$ (this is true since all the regions $S_i$ are cones with apex at the origin). This means we do not have to restrict to policies with $\sum t_i = 1$; we can choose a policy of any total time, as long as we normalize the utility by $\sum t_i$. This generalizes the strategy space, but is useful for the following reason. Consider a sequence of steps $\pi$ which starts at some point $P$ (not necessarily 0) and ends at $P$. Then if $U$ is the average utility of this cycle, then $U^* \ge U$ (in particular, we can consider any policy which goes from 0 to $P$ and then repeats this cycle many times). Likewise, if we have a sequence of steps $\pi$ which starts at some point $P$ and ends at $\lambda P$ for some $\lambda > 1$ which achieves average utility $U$, then again $U^* \ge U$ (by considering the policy which proceeds $0 \to P \to \lambda P \to \lambda^2 P \to \ldots$ (note that it is essential that $\lambda \ge 1$ to prevent this from converging back to 0 in finite time).

These observations motivate the following conjecture.

**Conjecture 10.** *The value $U^*$ is achieved by either:*

1. *The average utility of a policy starting at the origin and consisting of at most $N$ steps (in distinct regions).*

2. *The average utility of a path of at most $N$ steps (in distinct regions) which starts at some point $P$ and returns to $\lambda P$ for some $\lambda \ge 1$.*

We leave it as an interesting open problem to compute the optimal solution to this control problem.

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
