[Supplementary Material]

# Strategizing against No-regret Learners

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

))$ be a strategy for the control problem which satisfies $U(\pi) \geq U^* - 0.5\varepsilon$. As suggested by $\pi$, we will consider the strategy of the optimizer where for each $i$ (in order), the optimizer plays mixed strategy $\alpha_i$ for $t_i T$ rounds. We will show that this strategy guarantees an expected utility of $(U^* - \varepsilon)T - o(T)$ for the optimizer .

Since the learner is mean-based, they are playing a $\gamma$-mean-based algorithm for some $\gamma = o(1)$. As in Definition 2, let $\sigma_{j,t}$ be the learner's cumulative utility from playing action $b_j$ for rounds 1 through $t$. For $0 \leq i \leq k$, let $\tau_i = \sum_{j=1}^{i} t_i$ (with $T_0 = 0$). For $\tau \in [0, 1]$, let $P(\tau)$ be the state of the control problem at time $\tau$ (linearly interpolating between $P_i$ and $P_{i+1}$ if $\tau_i \leq \tau \leq \tau_{i+1}$); note that $P(\tau_i) = P_i$. We will first show that with high probability, $|\sigma_{j,\tau T} - T P(\tau)_j| \leq o(T)$; in other words, $P(\tau)$ provides a good approximation of the true cumulative utilities of the learner in the repeated game.

To see this, we first claim $|\mathbb{E}[\sigma_{j,\tau T}] - T P(\tau)_j| \leq k$. Fix any round $t$ in $[\tau_i T, \tau_{i+1} T]$; this means that the optimizer plays strategy $\alpha_i$ during round $t$, and therefore that $\mathbb{E}[\sigma_{j,t+1} - \sigma_{j,t}] = u_L(\alpha, b_j)$. If $t + 1$ also belongs to $[\tau_i T, \tau_{i+1} T]$ (so $t/T$ and $(t+1)/T$ both belong to $[\tau_i, \tau_{i+1}]$), we also have that $T(P(\frac{t+1}{T})_j - P(\frac{t}{T})_j) = u_L(\alpha, b_1)$. Since there are only $k$ intervals, $t$ and $t + 1$ belong to the same interval for all but $k$ rounds, and since utilities are bounded by 1 it follows that $|\mathbb{E}[\sigma_{j,\tau T}] - T P(\tau)_j| \leq k$. Now, we also claim that with high probability (at least $1 - 1/T$), for all $t$, $|\mathbb{E}[\sigma_{j,t}] - \sigma_{j,t}| \leq 10\sqrt{T \log(TN)}$. This follows simply from Hoeffding's inequality, since each component of $\sigma_{j,t}$ is the sum of $t$ independent random variables bounded in $[-1, 1]$. Together, this implies that $|\mathbb{E}[\sigma_{j,\tau T}] - T P(\tau)_j| \leq o(T)$.

We now claim that for sufficiently large $T$, the learner will play action $j_i$ for rounds $t \in [\tau_i T, \tau_{i+1} T]$. To see this, recall that $S_{j_i}$ is the unique region containing both $P_i$ and $P_{i+1}$. Since regions are convex with disjoint interiors, this means that the segment connecting $P_i$ and $P_{i+1}$ lies in the interior of $S_{j_i}$. By the definition of $S_{j_i}$, this implies that there exists some $\delta > 0$ such that for at least $1 - 0.5\varepsilon$ fraction of $\tau$ in the interval $[\tau_i, \tau_{i+1}]$, $P(\tau)$ satisfies $P(\tau)_{j_i} - P(\tau)_j \geq \delta$ for all $j \neq j_i$. Since $|\mathbb{E}[\sigma_{j,\tau T}] - T P(\tau)_j| \leq o(T)$ for all $j$, this means that for at least a $1 - 0.5\varepsilon$ fraction of rounds $t$ in $[\tau_i T, \tau_{i+1} T]$, we have that $\sigma_{j_i,\tau T} - \sigma_{j,\tau T} \geq \delta T - o(T)$. For sufficiently large $T$, this is bigger than $\gamma T$ (which is also $o(T)$).

Therefore, for each $i$, for at least $(1 - 0.5)\varepsilon t_i T$ rounds, the optimizer plays the mixed strategy $\alpha$ and the learner plays action $b_{j_i}$. The optimizer's total expected utility is therefore at least

$$\sum_{i=1}^{k} (1 - 0.5\varepsilon) t_i T u_A(\alpha_i, b_{j_i}) = (1 - 0.5\varepsilon) U(\pi) T \geq (1 - \varepsilon) U^* T.$$

## Part 2:

Assume there exists such a family (one for each $T$) of non-adaptive strategies $(\alpha^1, \alpha^2, \ldots, \alpha^T)$ for the optimizer . Since this strategy must work against any mean-based learner, we will construct a bad mean-based learner for this strategy in the following way. Fix $\gamma = T^{-1/2}$ (any $\gamma > 2/T$ will work). At any time $t$, let $J_t = \{b_j | \max_i \sigma_{i,t} - \sigma_{j,t} < \gamma T\}$ be the set of actions for the learner whose historical performance are within $\gamma T$ of the optimally performing action. The mean-based property requires the learner to play an action in $J_t$ with probability at least $1 - K\gamma$. Our mean-based learner will choose the *worst* action in $J_t$ for the optimizer ; that is, the action $b_j \in J_t$ which minimizes $u_A(\alpha^t, b_j)$.

Now, choose a sufficiently large $T_0$ such that this strategy achieves utility at least $(U^* + 0.5\varepsilon)$ for the optimizer against this mean-based learner. We now claim we can construct a solution $\pi$ to the control problem with $k = T_0$ which satisfies $U(\pi) \geq U^* + 0.5\varepsilon$, contradicting the optimality of $U^*$. Consider the protocol $\pi = ((\alpha^1, 1/T_0), (\alpha^2, 1/T_0), \ldots, (\alpha_0^T, 1/T_0))$. This is not a proper protocol, since some of the steps of this protocol might start in one region $S_j$ and end in a different region $S_{j'}$, but for any such steps we can divide them into substeps per region as described earlier.

420  We now claim that the step $(\alpha^t, 1/T_0)$ only passes through regions in the set $J_t$. To see this, note
421  that $P_t$ and $P_{t+1}$ differ in each coordinate by at most $1/T_0$ (since all utilities are bounded by 1).
422  Therefore if the segment between $P_t$ and $P_{t+1}$ passes through a point on the boundary $S_j \cap S_{j'}$
423  (where $u_j = u'_j = \max_i u_i$), it must be the case that $(P_t)_j$ and $(P_t)_{j'}$ are both within $2/T_0$ of
424  $\max_j (P_t)_j$. By construction $(P_t)_j = \frac{1}{T_0}\sigma_{j,t}$, so this implies that $\max_i \sigma_{i,t} - \sigma_{j,t} \leq 2 \leq \gamma T$, and
425  therefore $j \in J_t$ (similarly, $j' \in J_t$).

426  Now, if the step $(\alpha^t, 1/T_0)$ only passes through regions in the set $J_t$, it obtains utility for the optimizer
427  at least $\min_{b_j \in J_t} \frac{1}{T_0} u_A(\alpha^t, b_j)$, and thus

$$U(\pi) = \frac{1}{T_0} \sum_t \min_{b_j \in J_t} u_A(\alpha^t, b_j).$$

428  But this sum is exactly the utility of the optimizer against our mean-based learner, which is at least
429  $(U^* + 0.5\varepsilon)T_0$. It follows that $U(\pi) \geq U^* + 0.5\varepsilon$, contradicting that $U^*$ is optimal.

430  $\qquad\qquad\qquad\qquad\qquad\qquad\qquad\qquad\qquad\qquad\qquad\qquad\qquad\qquad\qquad\qquad\qquad\qquad\qquad\qquad\qquad$ $\square$