[Reviews · NeurIPS 2019]

Reviewer 1



This paper asks how a player should exploit knowledge that their opponent in a repeated game is using a no-regret learning algorithm. Prior work has studied this question in Bayesian settings, such as when the learning player is a buyer and the rational player is a seller. This question extends the ideas to a non-Bayesian setting. In general, the rational player can guarantee the first-mover Stackelberg utility in the game. That is, being rational against a no-regret learner is worth at least as much as going first in a Stackelberg game. If the learning is using a "mean-based" learning method, the rational player can obtain even more utility. I like this paper. The question of how to strategize vs a learning agent is a very hot topic. The result that the rational player can guarantee their Stackelberg utility is quite easy to prove, but is a nice conceptual result none-the-less. The authors also show that certain learning guarantees (no-swap-regret) prevent the rational player from getting more than this, which says something interesting about the use of no-swap-regret methods in competitive environments. The result about improving on Stackelberg utility against mean-based learners is very similar to Braverman et al (2018), but applying these ideas in the non-Bayesian setting is a solid marginal contribution. Overall, this paper makes a solid conceptual contribution in a hot area. The marginal technical contributions are not especially deep, but on net the paper would be a good fit for NeurIPS. I think the paper is a good candidate for acceptance.

Reviewer 2



Summary of paper: Considers strategies for playing a repeated, general two-player game against an online learning algorithm. Shows tight connections to the Stackelberg equilibria of the game. In general one can guarantee to achieve the utility of being the leader in a Stackelberg equilibrium, and this is tight if the learner has no swap regret. One can gain more against standard "mean-based" learners like FTRL. Summary of opinion: I think this paper asks a very interesting and exciting question in the field of online learning and game-playing (which so far mostly focuses on zero-sum games or specific games like auctions), and provides a useful and valuable solution. I found the paper clear and well-written. The connection to Stackelberg equilibrium is intuitive and clear in hindsight, but an important step forward for the literature. Originality: strong as far as I know. Generalizes some very specific cases in the literature (auctions). Clarity: very good in my opinion. Significance: Quite high. No-regret learning is a broad area of high interest, and I think many researchers will learn from these results and want to build on them. Suggestions for writing clarity: Line 34 - what is epsilon? Definition 2 - asymptotics of gamma are a bit confusing, is gamma a function of T, or t, or neither? Please respond to any part of my review that you feel appropriate.

Reviewer 3



Braverman et al in EC'18 considered how to sell items to a no-regret buyer. The current paper offers a broad generalization of this result (in a bit of a different context, as buying items in a Bayesian game, while they consider general matrix games). The results are as follows 1. against a no-regret learner one can guarantee at least the Stackelberg value of the game (that is: the value one can get by committing to a move first, and letting the opponent best respond). If the learner is using a no-swap-regret strategy, than this is *exactly* the value attainable (no matter what that strategy is). I find this the most important/interesting finding of the paper. In contrast, if the learner is using a mean-based learning strategy, than an example is provided to show that one can get higher value against such a learner, and theorem 9 offers a characterization of the optimal strategy against such a player. The main result of the Braverman et al paper in EC'18 is an analogous result for auction games, showing that the auctioneer can extract more than the optimal revenue of the 1-shot game against such mean-based learners. I have read the author response. It is unfortunate that the paper does not extend the work to Bayesian games, and hence doesn't offer a clear comparison with the Braverman paper. But the results are very strong, and enlightening.

[Author Response · NeurIPS 2019]

We thank all the reviewers for their positive and encouraging feedback. We agree that understanding the solution to the control problem better and extending these results to Bayesian games are interesting avenues of research and are currently working in these directions.

[Meta-Review · NeurIPS 2019]

All reviews are very positive. The paper improves over [BMSW] in a very interesting way, and forms an interesting interplay/complimentarity with [BMSW]. In particular, it presents a clear and tight characterization of what the learner's no-regret property implies for the opponent's best behavior. [BMSW] Mark Braverman, Jieming Mao, Jon Schneider, and Matt Weinberg. Selling to a no-regret buyer. EC'18.